# Associations between Feeding Behaviors Collected from an Automated Milk Feeder and Neonatal Calf Diarrhea in Group Housed Dairy Calves: A Case-Control Study

**DOI:** 10.3390/ani12020170

**Published:** 2022-01-11

**Authors:** Meridith H. Conboy, Charlotte B. Winder, Melissa C. Cantor, Joao H. C. Costa, Michael A. Steele, Catalina Medrano-Galarza, Taika E. von Konigslow, Amanda Kerr, Dave L. Renaud

**Affiliations:** 1Department of Population Medicine, University of Guelph, Guelph, ON N1G 2W1, Canada; mconboy@uoguelph.ca (M.H.C.); winderc@uoguelph.ca (C.B.W.); cantorm@uoguelph.ca (M.C.C.); cata.medrano@outlook.com (C.M.-G.); tvonkoni@uoguelph.ca (T.E.v.K.); 2Department of Animal and Food Sciences, University of Kentucky, Lexington, KY 40508, USA; costa@uky.edu; 3Department of Animal Biosciences, University of Guelph, Guelph, ON N1G 2W1, Canada; masteele@uoguelph.ca; 4Programa de Maestría en Bienestar Animal, Facultad de Medicina Veterinaria, Universidad Antonio Nariño, Bogota 110311, Colombia; 5Grober Nutrition, Cambridge, ON N1T 1S4, Canada; akerr@grober.com

**Keywords:** calf health, group housing, automated, precision livestock farming, precision dairy

## Abstract

**Simple Summary:**

Precision technology devices are often integrated on dairies to monitor animal health. One precision technology used to manage calves is an automated milk feeder which can record feeding behaviors such as daily milk intake, drinking speed, and feeder visits. The objective of this study was to determine if calf feeding behaviors collected by an automated milk feeder, changed in the days surrounding diagnosis of neonatal calf diarrhea (NCD; e.g., −3 days to 5 days after diagnosis). Milk intake was lower for the greatest number of days surrounding diagnosis of NCD compared to control calves, but the sensitivity and specificity of detecting NCD using any individual behavior was low. However, parallel testing using cumulative feeding behaviors on the day of diagnosis of NCD increased the sensitivity for detecting disease. This study provides insights into the association of feeding behavior with calves destined for NCD using an automated milk feeder. We suggest feeding behaviors cannot replace visual diagnosis of NCD, but that feeding behaviors might serve as a screening tool for producers.

**Abstract:**

The objective of this case-control study was to determine if feeding behavior data collected from an automated milk feeder (AMF) could be used to predict neonatal calf diarrhea (NCD) in the days surrounding diagnosis in pre-weaned group housed dairy calves. Data were collected from two research farms in Ontario between 2017 and 2020 where calves fed using an AMF were health scored daily and feeding behavior data (milk intake (mL/d), drinking speed (mL/min), number of rewarded or unrewarded visits) was collected. Calves with NCD were pair matched to healthy controls (31 pairs) by farm, gender, and age at case diagnosis to assess for differences in feeding behavior between case and control calves. Calves were first diagnosed with NCD on day 0, and a NCD case was defined as calves with a fecal score of ≥2 for 2 consecutive days, where control calves remained healthy. Repeated measure mixed linear regression models were used to determine if there were differences between case and control calves in their daily AMF feeding behavior data in the days surrounding diagnosis of NCD (−3 to +5 days). Calves with NCD consumed less milk on day 0, day 1, day 3, day 4 and day 5 following diagnosis compared to control calves. Calves with NCD also had fewer rewarded visits to the AMF on day −1, and day 0 compared to control calves. However, while there was a NCD status x day interaction for unrewarded visits, there was only a tendency for differences between NCD and control calves on day 0. In this study, feeding behaviors were not clinically useful to make diagnosis of NCD due to insufficient diagnostic ability. However, feeding behaviors are a useful screening tool for producers to identify calves requiring further attention.

## 1. Introduction

Automated milk feeders (AMF) are increasing in popularity [1] and promote calf welfare by encouraging natural feeding behavior patterns [2]. Furthermore, AMF are a social system, and calf companionship has been associated with reduced neophobia towards novel environments [3] and novel feeds [4]. However, concerns over the horizontal spread of disease between calves has led some producers to hesitate towards adopting automated milk feeding systems [5]. For example, group housing was associated with a higher prevalence of bovine respiratory disease (BRD) in calves compared to individual housing [6,7]. In contrast, the incidence of diarrhea was found to be lower in groups of four calves compared to individual housing [8]. However, it was recently found that management factors were associated with morbidity rates [9] and mortality rates [10] for farms feeding calves with automated milk feeding systems. For example, bacterial contamination of the milk fed to calves [9] and higher stocking densities [11] were associated with a higher likelihood of some signs of BRD for AMF calves. Furthermore, it has been observed that the percentage of calves with neonatal calf diarrhea (NCD) and BRD more than doubled from 20% (e.g., all-in-all-out practices) to over 40% (e.g., dynamic group housing) when housing practices changed in the same AMF facilities [12]. Thus, it is likely that management factors are responsible for increased disease transmission in calves rather than automated milk feeding systems per se.

One advantage to AMF is their ability to collect calf feeding behaviors such as total daily milk intake (mL/day), drinking speed (mL/min), number of rewarded visits (calf receives milk) and number of unrewarded visits (non-nutritive visits) to the feeder. Since sickness behavior leads to reduced appetite in mammals [13], there is the potential to use AMF for detecting NCD on farm. For example, when calves were offered at least 8 L/d, decreased drinking speed, unrewarded visits, and milk intake were associated with NCD status in the days prior to and after producer diagnosis compared to healthy calves [14]. Similar results were observed with calves offered 6 L/d and observed by trained researchers, calf milk intake was lower than healthy calves up to four days prior to NCD, and unrewarded visits were lower up to two days before disease diagnosis [15]. Finally, calves offered a high milk allowance of 12 L/d had less milk intake, less total feeder visits, and occupied the feeder stall for less time than healthy calves following NCD diagnosis by a veterinarian [16]. Moreover, calves in a cross-sectional study consumed less milk, and had slower drinking speeds than healthy calves on the day of NCD diagnosis [17]. These studies suggest that feeding behaviors are likely associated with NCD status in calves. However, as reported in a scoping review, very few studies have used abnormal fecal consistency scoring as their diagnostic criteria [18], and this is imperative since it is one of the only validated metrics for diagnosing diarrhea on farm [19,20]. Furthermore, since NCD development occurs quickly [21,22], and many studies only health scored calves biweekly, it is imperative that calves are health scored by trained researchers daily. Thus, research is needed which evaluates the association of feeding behaviors with NCD in calves using a validated fecal consistency scoring system.

The objective of this case-control study was to determine if feeding behavior was associated with NCD status in preweaned calves in the three days prior to and the five days after NCD diagnosis when compared to healthy controls. It was hypothesized that milk intake, drinking speed, and unrewarded visits would decrease in the days surrounding NCD diagnosis when compared to healthy calves.

## 2. Materials and Methods

This case-control study was reported using the STROBE-Vet checklist for observational studies [23] and was approved by the University of Guelph Animal Care Committee (Animal Use Protocols #3477 and #4408). Data were collected from January to July 2017 and January to November of 2018 at the University of Guelph Livestock Research and Innovation Centre—Elora Dairy Facility (Elora, ON, Canada) and from June to August of 2020 at the Grober Young Animal Development Center (Cambridge, ON, Canada). These data were collected from 174 Holstein calves, the majority of which were female, as part of two other studies, one of which was published [24]. The two research facilities differed in that the Elora Dairy Facility was a closed herd whereas the Grober Young Animal Development Center was an open herd as they bought calves of varying age and sex from various farms. In brief, calves at the Elora Dairy Facility were fed 3 L of colostrum by bottle within 2 h and offered another 3 L 12 h later [24]. Since Grober Young Animal Development Center was an open herd, colostrum management information for these calves was not available. For this study, a total of 62 calves were enrolled (31 cases and 31 controls), allowing for the detection of a significant difference in drinking speed with 95% confidence and 80% power. This was calculated using the mean drinking speed and standard deviation of healthy calves (896 ± 337 mL/min) and diseased calves (645 ± 345 mL/min; [14]).

### 2.1. Calf Feeding Protocol

All calves were fed via a Förster Technik (Cambridge, ON, Canada) AMF using the machine standard 40 FIT feeding plan which provided calves with milk ad libitum (maximum of 3 L per 2 h) for the first 35 days on the AMF (preweaning period), while calves were weaned from 12 to 2 L of milk replacer in the remaining 35 days. The milk replacer was formulated to have 26% CP and 18% crude fat (Excel Pro-Gro, Grober Nutrition, Cambridge, ON, Canada) and was fed at a concentration of 150 g/L at both facilities. At the Elora Dairy Facility, calves were housed in dynamic groups of 10 to 15 calves per pen with a lying surface of 2.8 m^2^ per calf. At the Grober Young Animal Development Center, calves were housed in dynamic groups of 7 calves per pen with a lying surface of approximately 4.1 m^2^ per calf. The bedding material was the same at both facilities and consisted of wood shavings on cement. Soiled bedding was removed daily, and the bedding was removed and replaced weekly. Calves at the Elora Dairy Facility were enrolled at birth and health scored until weaning while calves at the Grober Young Animal Development Center were enrolled at arrival to the facility and health scored daily for three weeks. The mean age of the calves at enrollment was 8 days (range: 3 to 21 days; Interquartile Range (IQR): 5 to 10 days).

### 2.2. Calf Health Measurements and Interventions

Calves were health scored daily using a fecal consistency scoring system [19]. The rectum of the calves was palpated with a rectal thermometer to initiate defecation. Calves were scored on a 4-point scale with 0 being normal (firm), 1 being soft (moderate spreading), 2 being runny (substantial spreading) and 3 being severely abnormal and watery (splatters and sifts through bedding) [19]. Calves at the Elora Dairy Facility were health scored from birth until the end of the weaning period, whereas calves at the Grober Young Animal Development Center were health scored for 21 consecutive days, starting on the day after arrival to the facility. A total of 4 observers performed the daily health scoring exams. The observers were trained by a veterinarian or were a veterinarian. No formal intra-observer reliability testing was conducted.

A mild case of NCD was defined as a calf who had abnormal fecal consistency ≥2 for at least one day. Electrolyte treatments for mild cases of NCD were the same for both facilities. Mild cases of NCD were treated with oral electrolytes (Calf-Lyte, Vetoquinol, Lavaltrie, QC, Canada) twice daily with the Elora Dairy Facility also administering meloxicam (Metacam [2.5 mL per 100 kg s.c.] Boehringer Ingelhein, Burlington, ON, Canada) once on the day of diagnosis. Calves were treated with oral electrolytes via a bottle, or esophageal tube if they refused the bottle, until they consumed at least 6 L/d of milk replacer. A severe case of NCD was defined as a calf who was depressed and required oral electrolytes for three or more days. Severe cases of NCD at Grober Young Animal Development Center were treated with electrolytes 3 times daily as needed thereafter, trimethoprim-sulfadoxine (Trimidox [3 mL per 45 kg body weight i.m. once daily for 3 consecutive days], Vetoquinol, Lavaltrie, QC, Canada) and with meloxicam (Metacam [2.5 mL per 100 kg s.c.] Boehringer Ingelhein, Burlington, ON, Canada). Calves with severe cases at Elora dairy facility were treated similarly to those at the Grober Young Animal Development Center, however, calves were also provided with intravenous rehydration therapy (Electrolyte Infusion and Physiological Saline, Bimeda, MTC Animal Health Inc., Cambridge, ON, Canada), at the discretion of the attending veterinarian, when the calf was dehydrated and did not voluntarily consume milk replacer or electrolytes.

### 2.3. Case and Control Selection

Calves 35 days or older were excluded from this study as they would have already entered the weaning process with restricted milk allowances, and there is variability in feeding during weaning in calves [25] We also required the feeding behavior data to be complete for all calves enrolled on this study from days −3 to day 5 after NCD diagnosis. Of the 174 calves evaluated, 31 calves met the criteria to be a case and 31 calves were selected to be the controls.

Cases were defined as having NCD if they had a fecal score of 2 or 3 for a minimum of 2 consecutive days. The second day of having a fecal consistency score of 2 or 3 was defined as the day of diagnosis (day 0). Control calves were defined as calves who did not have an abnormal fecal consistency score for two consecutive days. Case and control calves were matched by farm, gender, and age. Case and control calves were matched as close as possible on age, with the maximum age difference between them being 7 days. To account for lack of independence, calves that were from the two different trials conducted at the Elora Dairy Facility were from two different farms to account for year and any potentially different environmental or management practices between the different years the data was collected.

### 2.4. Feeding Behavior Measurements

Drinking behaviors [milk intake (mL/d), drinking speed (mL/min), number of rewarded visits per day and the number of unrewarded visits per day] of the calves were collected from the AMF throughout the observation period. To do so, the Microsoft Excel files with all the feeding behavior data was collected directly from the AMF using an SD card. The milk intake, number of rewarded visits and number of unrewarded visits for each visit were cumulated to provide a total amount for each day. The drinking speed from each visit in the day was averaged to provide one value for the day. For the purposes of this study, data were evaluated from 3 days prior to disease diagnosis (day −3) until 5 days after disease diagnosis (day 5) where the day of diagnosis was denoted as day 0. Previous work identified that feeding behaviors changed in the ±10 d surrounding disease diagnosis [14]; however, in order to have a sufficient sample size, our study’s observation period was limited to day −3 to day +5 due to cases exhibiting abnormal fecal consistency scores early in life.

### 2.5. Statistical Analysis

All data were recorded in Microsoft Excel (Windows 10, Microsoft Corp., Redmond, WA, USA). Mixed linear models (Proc Mixed) were analyzed in SAS 9.4 (SAS, Cary, NC, USA) and each drinking behavior outcome (milk intake, drinking speed, rewarded visits, and unrewarded visits) were run separately with day relative to NCD diagnosis as a repeated measure and calf as the subject during the observation period (days −3 to 5 days after NCD diagnosis, referred to in results as the observation period). In each model, pair and NCD status were offered as fixed effects, and day relative to NCD diagnosis (day −3 before NCD diagnosis to day 5), the NCD status x day relative to NCD diagnosis interaction and the total number of days on the AMF were offered as covariates.

Multicollinearity was assessed for all variables, where a correlation coefficient of 70% or greater indicated collinearity; however, no variables were considered collinear. Unstructured, Toeplitz, exchangeable, and first order autoregressive covariance structures were tested for each model, and the AIC value was determined. The covariance structure that offered the lowest AIC value was used to analyze the data as it best explained the correlation within the data. The Toeplitz covariance structure was used for milk intake, drinking speed, and rewarded visits. The unstructured covariance structure was used for unrewarded visits.

Univariable analysis was conducted using a liberal *p* value of 0.20 for the predictor variable and all covariates. The final multivariable model was obtained through backwards stepwise elimination to ensure all variables included in the model had a *p* ≤ 0.05 unless it was a confounder or part of a significant interaction term. If a non-significant variable was removed from the model and the difference in the coefficients of a significant variable changed by ≥25%, the variable was deemed a confounder and was retained in the model. If the interaction term between day relative to diagnosis and case of NCD was found to be significant, it was kept in the model and evaluated further through contrast tables, predictive margins, and margins plots.

Normal quantile plots were used to assess normality of the residuals for each dependent variable. Unrewarded visits were not normally distributed and were transformed with common log and a correction factor of 1. The back-transformed least square means minus the correction factors and the 95% CI are reported for unrewarded visits, with statistical significance reported on the modeled transformed values.

Outliers were assessed graphically and were further scrutinized if the standardized residuals were >3 or <−3. The raw data was assessed to ensure that all outliers were biologically plausible, and only one observation was detected as biologically impossible in the drinking speed model and was removed. Models of milk intake, drinking speed, rewarded visit and unrewarded visits had 5, 2, 6, and 15 outliers, respectively. All outliers were assessed for model leverage using the influence option in Proc Mixed which also calculated Cook’s Distance, and restricted likelihood distance. These outliers did not have high leverage on the models, so they were retained in the models.

To determine the cut point which optimized sensitivity and specificity for each outcome, Youden’s Index was used on the day of NCD diagnosis. Sensitivity was the defined as the probability that a truly diseased calf is identified as diseased based off its feeding behaviors collected from the AMF, whereas specificity was defined as the probability that a truly healthy calf was identified as being healthy based off its feeding behaviors collected from the AMF. Parallel testing, analyzed in Stata 15 (Stata Corp., College Station, TX, USA), was conducted on the day of diagnosis to determine if the sensitivity of disease prediction could be improved. All feeding behaviors that were found to be statistically significant in the mixed models (on the day of diagnosis) were assessed cumulatively. If any of the significant feeding behaviors were below the optimal cut point of disease detection on the day of diagnosis, the calf was considered diseased. The sensitivity and specificity were then recalculated using the results of parallel testing.

## 3. Results

A total of 174 calves were evaluated. Of these, 31 pairs (62 calves of which 16 were bulls and 46 were heifers) met the inclusion criteria and were selected from the farms. There were 20 pairs of calves from the Elora Dairy Facility (17 pairs from 2017 and 3 pairs from 2018) and 11 pairs of calves from the Grober Young Animal Development Center. Median age at diagnosis (day 0) was 11 days of age (range: 6 to 24 days; interquartile range (IQR): 8 to 14 days) while median days on the AMF at diagnosis was 7 days (range: 4 to 16 days; IQR: 6 to 8 days). Baseline characteristics of case and control calves are presented in Table 1.

### 3.1. Association of Feeding Behavior for the Observation Period

The association of feeding behavior and NCD status, and the NCD status x day interactions for the observation period are presented in Table 2. Briefly, milk consumption was lower for NCD calves compared to control calves *(p* < 0.001). There was also an NCD status x day interaction for rewarded visits *(p* < 0.001), unrewarded visits *(p* < 0.001), and there was a tendency for a NCD status x day interaction for drinking speed (*p* = 0.07).

### 3.2. Milk Intake

There was a negative association of milk intake with NCD status (*p* < 0.001), and there was a significant NCD status x day interaction for milk intake (*p* < 0.001; Figure 1). Specifically, NCD calves drank less milk than control calves for several days (day 0, day 1, day 3; *p* < 0.001, day 4; *p* = 0.03, and day 5; *p* = 0.02). Neonatal calf diarrhea calves also tended to drink less milk on days −1 (*p* = 0.09) and day 2 (*p* = 0.06) compared to control calves. Day −3 and day −2 were not different (*p* > 0.10).

### 3.3. Drinking Speed

There was a tendency for a NCD status x day interaction for drinking speed (*p* = 0.07; Figure 2). Specifically, there was a tendency for NCD calves to have a slower drinking speed than control calves on day 1 (*p =* 0.09), but no other days were different (days −3 to day 0 and days 2 to day 5; *p* > 0.10).

### 3.4. Rewarded Visits

There was a NCD status x day interaction for rewarded visits (*p* < 0.03; Figure 3). Specifically, NCD calves had less rewarded visits surrounding NCD diagnosis (day −1; *p* = 0.02 and day 0; *p* = 0.04). However, NCD calves had greater rewarded visits on day 3 (*p* = 0.03) and tended to have greater rewarded visits on day 4 (*p* = 0.09) compared to control calves, but no other days were different (days −3 to day −2 and day 1, day 2, and day 5; *p* > 0.10).

### 3.5. Unrewarded Visits

There was a NCD status x day interaction for unrewarded visits (*p* = 0.001; Figure 4). Specifically, NCD calves tended to have less unrewarded visits on the day of NCD diagnosis (day 0; *p* = 0.07) compared to control calves. However, there were no other differences by day observed between NCD and control calves (days −3 to day −1 and days 1 to day 5; *p* > 0.10).

### 3.6. Parallel Testing

Dependent variables which had a significant interaction for NCD status x day and at least one significant day difference between control and NCD calves were used to calculate parallel sensitivity and specificity on the day of diagnosis, as this was when most feeding behaviors had a greater effect and were statistically significant. This criteria were met by two dependent variables, milk intake and rewarded visits. The optimal cut points used in this calculation can be found in Table 3**.** Parallel interpretation of the feeding behavior cut points resulted in a sensitivity of 69% and a specificity of 22%.

## 4. Discussion

Our study provides insight into how feeding behaviors change surrounding a diagnosis of NCD in calves. A significant decrease in milk intake was found in calves with NCD compared to control calves. Furthermore, there was a NCD status x day interaction for rewarded visits, which were less for NCD calves on day −1 and day 0, but greater on day 3 compared to control calves. Milk intake has been associated with NCD in other studies as highlighted by a scoping review [18]. Our findings highlight that the onset of NCD in calves leads to sickness behavior and anorexia which are the first behavioral changes that occur when the immune response is activated [13]. However, this study differs in that the calves were health scored daily using a validated fecal consistency scoring system for indicating diarrhea in calves by trained researchers. This is different than others who either used producer diagnostic records [14], assessed the calf hide for dirtiness or severe diarrhea [15], or only evaluated for associations of feeding behavior with NCD on the day of diagnosis [17], which increased our detection ability of NCD and allowed for changes in feeding behaviour to be detected as soon as symptoms of disease were present. However, cumulative parallel testing which included rewarded visits and milk intake did not achieve a high enough sensitivity to replace visual diagnostic criteria. Thus, we suggest that feeding behaviors may be useful to identifying a calf requiring further examination, but changes in these behaviors cannot indicate NCD status alone.

### 4.1. Associations between Feeding Behaviors and Neonatal Calf Diarrhea

We observed that milk intake was the feeding behavior affected for the most days during NCD diagnosis and afterwards. Others also observed that milk intake declined on the day of NCD and after diagnosis in calves [14]. However, we only observed a tendency for differences prior to NCD diagnosis on day −1, which disagreed with a cross-sectional study [14] and an observational study [15]. It is possible that differences were due to different diagnostic methods used for NCD. For example, the cross-sectional study relied on producer reporting, a sensitivity of 26%, specificity 97% and 84% accuracy was reported for diagnosing calves with a health event compared to a veterinarian [14]. This could have led to calves with NCD not being identified by the calf caregiver. As a result, NCD calves on our study may have been identified earlier, suggesting that the day relative to NCD diagnosis may not be directly comparable between their study and our study. Furthermore, an observational study observed that calves with NCD had consumed significantly less milk than controls on days −4, day −2, day −1 and day 0 compared to control calves [15]. While we also observed in this study that our calves had less milk intake prior to NCD diagnosis when compared to control calves, the other study diagnosed calves as NCD positive if they were observed to have loose or watery feces and a fever (rectal temperature of 39.5 °C or more), or if hide dirtiness was observed [15]. Recently, hide dirtiness score upon arrival to a veal farm was found to be a poor indicator of NCD in calves [20]. Thus, more severe cases of NCD, plus the inclusion of calves with hide dirtiness to diagnose NCD explains differences between our studies, making comparisons difficult.

Drinking speed was found to not differ between case and control calves in this study, which disagreed with a previous cross-sectional study, where drinking speed was different from day −3 to day 10 after NCD diagnosis [14]. As mentioned previously, this difference is likely due to differences in disease measurement, as our study likely detected a calf with an abnormal fecal consistency score sooner. In addition, calves in our study consumed milk much slower than what was observed in that study [14; 490 ± 218 mL/min vs. 877 mL/min ± 344 mL/min, respectively], and was likely due to differences in the milk feeding strategy programmed into the automated milk feeder. Drinking speed has been shown to vary based on milk feeding strategies, calves offered less milk (e.g., <8 L/d) had faster drinking speeds than calves offered more milk [26]. Indeed, this hypothesis agreed with the findings of an observational study, drinking speed was not different between sick and healthy calves when two different milk feeding strategies were used [27], which could have greatly impacted the feeding behaviors. Thus, we suspect that drinking speed is dependent on milk feeding strategy and NCD definitions but was not a reliable indicator of NCD in this study.

In this study, the number of rewarded visits to the AMF per day was different between diseased calves and control calves on day −1, day 0, and day 3, which has not been observed in other studies [14,15,27]. Indeed, no differences in the number of rewarded visits between control calves and calves with NCD were observed in other studies [27]. The differences between the studies and our study are likely in part due to the method of health scoring and definition of disease as previously discussed. However, the most likely cause of this difference is milk feeding strategy. Calves in our study were offered milk ad libitum, while other studies offered calves 8 L/d or less [14,15]. It has been observed that milk feeding strategy affects how often a calf visits the feeder [28]. For example, AMF calves offered milk ad libitum consumed twice as much milk (e.g., 12 L/d), had minimal unrewarded visits, and had greater rewarded visits compared to calves offered restricted milk at 10% body weight per day [28]. Thus, it is possible that NCD calves in the aforementioned studies had a ceiling effect for rewarded visits, where the maximum milk allotment offered was less than the total amount of milk that a calf wanted to consume. We suggest that rewarded visits may change in calves offered milk ad libitum, but our results cannot replace a diagnostic calf exam.

In this study, we observed a tendency for unrewarded visits to be different on the day of NCD diagnosis. Unrewarded visits have been considered an indicator of hunger, as calves offered restricted amounts of milk have higher unrewarded visits than calves offered milk ad libitum [28]. Indeed, other studies which restricted milk intake observed an association between unrewarded visits and NCD status in calves [14,15]. We suspect differences in our findings are because our calves always had milk available, and there were minimal opportunities to experience an unrewarded visit since calves could consume up to 3 L every 2 h. Alternatively, unrewarded visits were not-normally distributed and had to be transformed in this study to assess for significance, we suggest that this variable is highly variable by individual calf. It is possible that for calves who are milk motivated, changes in unrewarded visits might be indicative of NCD, personality traits have been associated with feeding behavior in calves [25]. Future research should investigate the potential of changes in an individual calf’s unrewarded visits to predict neonatal calf diarrhea to determine if unrewarded visits can be used as an indicator of neonatal calf diarrhea.

### 4.2. Parallel Testing

The findings in our study suggest that feeding behaviors cannot be used as the sole indicators of diagnosis of NCD in calves. Thus, we used cumulative testing, where we evaluated the potential of behavioral changes in milk intake and rewarded visits to indicate disease. While we observed a moderate sensitivity, we observed a low specificity for diagnosing NCD in these calves. As was suggested by epidemiologists decades ago, a high sensitivity rate is desired when the consequences of the disease can be fatal [29]. Since NCD is the leading cause of morbidity and mortality in preweaned calves [30], we suggest that the use of these feeding behaviors to identify NCD cannot replace physical exams. However, feeding behavior may be useful for identifying which calves require further attention since some calves with NCD require immediate intervention, calves with NCD can present signs of dehydration, metabolic acidosis, hypoglycemia and hyperatremia [31,32], which require timely clinical interventions such as fluid therapy to rectify [33,34]. Future research should develop an alert using feeding behavior to detect calves at risk for neonatal calf diarrhea.

### 4.3. Limitations

A limitation to consider when interpreting the results of our study is the short period calves were observed. Others have found that some feeding behaviors changed in the days leading up to diagnosis and then persisted throughout the entire study period [14,15]. As a result, ideally the observation period would have been larger than what was used in our study; however, many calves had NCD soon after they were trained on the AMF, limiting the amount of feeding behavior data that was available for them prior to diagnosis. Thus, we were unable to assess for changes in feeding behaviors for 4 or more days prior to diagnosis as there would have been an insufficient sample size and as a result, it is unknown if there were changes in feeding behaviors in those days. Furthermore, this data was collected as part of a convenience sample which may have reduced the external validity of the study. For example, since calves from the Grober facility were sourced from multiple locations, we did not have serum total protein status or birthweights to include as a covariate in the models. In addition, bias may have also been introduced into the study as the calves were treated with oral electrolytes and meloxicam at the onset of neonatal calf diarrhea. Providing an oral electrolyte solution to NCD calves has been associated with improved hydration status, improved blood pH, and a lower duration to resuscitation compared to calves receiving fluid therapy IV or subcutaneously [34]. It is possible that providing an oral electrolyte solution to calves in this study may have influenced calf feeding behaviour compared to calves not receiving a treatment. However, due to the virulence of NCD, oral electrolytes were provided to the calves to minimize suffering and the duration of disease. Furthermore, it has been shown that administering meloxicam to calves with NCD increased the likelihood of the calf consuming the entirety of their milk allowance [35]. However, the effect of NSAID on milk intake has yet to be investigated in calves offered more than 4 L/d. A final limitation to consider is that multiple researchers completed health scoring of calves and no interobserver reliability tests were completed. This may have resulted in differences in the fecal consistency scoring, and thereby misclassification of disease status.

In summary, we suggest that milk intake and rewarded visits are associated with neonatal calf diarrhea in calves offered milk ad libitum, with milk intake being the most robust behavior for detecting changes associated with NCD; however, we caution that this observation is dependent on the milk feeding strategy programmed into the automated milk feeder. Based on the literature, we believe that more complex algorithms are needed to increase the sensitivity and specificity of disease detection for NCD in calves, because the time between the pathogen exposure, incubation period and symptoms of NCD is around 24–48 h in calves [21,22]. It is possible that multiple behaviors (i.e., activity levels with feeding behavior) may increase the sensitivity and specificity of an algorithm’s ability to detect disease in calves in a timely manner. Alternatively, it is also possible that with a larger population of dairy calves, machine learning techniques can tease out feeding behavior differences associated with NCD without incorporating multiple precision technologies. Thus, more research is needed to develop methods which increase the sensitivity and the specificity of disease detection using feeding behavior in calves.

## 5. Conclusions

This case-control study indicated that calves with neonatal calf diarrhea changed their feeding behaviors in the days surrounding the diagnosis of disease. A matched pair analysis revealed that calves with neonatal calf diarrhea had less rewarded visits on the day before diagnosis, and the day of diagnosis of disease; lower milk intake was also observed for these calves on the day of disease diagnosis and for several days after disease diagnosis compared to control calves. The automated milk feeder cannot replace the clinical examination used to identify calves with neonatal calf diarrhea. Rather, it can be used as tool to help indicate which calves may require further attention. Future research should target larger sample sizes and observational periods to further assess the associations between disease and changes in feeding behaviors in calves. Future research should also evaluate the potential of feeding behavior algorithms for an early intervention strategy to ameliorate neonatal calf diarrhea.

## Figures and Tables

**Figure 1 animals-12-00170-f001:**
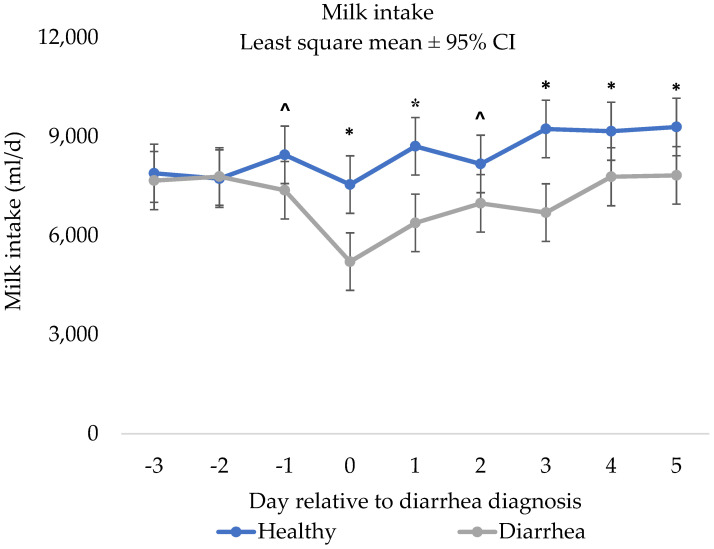
Results of the repeated measures mixed linear regression model assessing the relationship between day relative to neonatal calf diarrhea diagnosis (day 0) and least square mean milk intake (mL/d) of pair matched control (*n* = 31) and diarrhea (*n* = 31) calves fed by an automated milk feeder, preweaning. A calf was defined as having neonatal calf diarrhea if they had a fecal consistency score of 2 or 3 (runny to watery consistency) for two or more consecutive days. Calves were matched on farm, gender and age and the model included calf and pair number to account for the matched pair analysis. The error bars represent ±95% confidence interval. * Indicates a significant difference (*p* ≤ 0.05), and ^ indicates a tendency (> 0.05 *p* < 0.10).

**Figure 2 animals-12-00170-f002:**
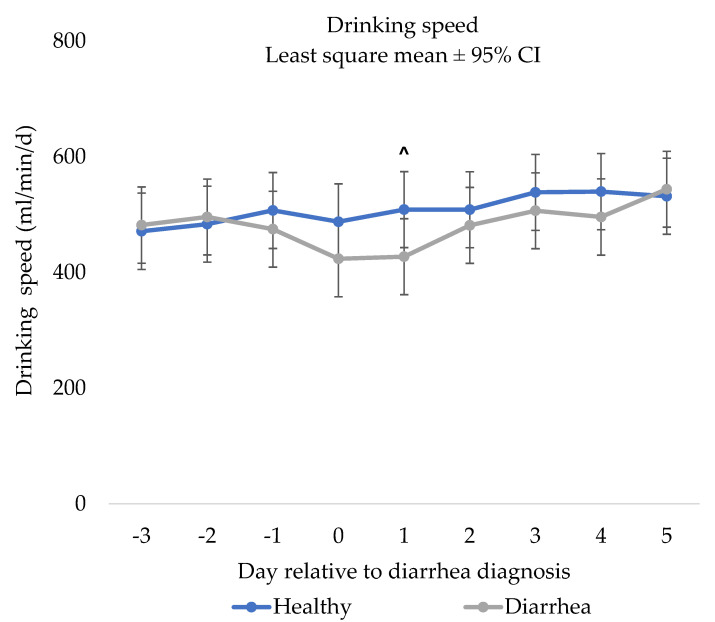
Results of the repeated measures mixed linear regression model assessing the relationship between day relative to neonatal calf diarrhea diagnosis (day 0) and drinking speed (mL/d) of pair matched control (*n* = 31) and diarrhea (*n* = 31) calves fed by an automated milk feeder, preweaning. A calf was defined as having neonatal calf diarrhea if they had a fecal consistency score of 2 or 3 (runny to watery consistency) for two or more consecutive days. Calves were matched on farm, gender and age and the model included calf and pair number to account for the matched pair analysis. The error bars represent ±95% confidence interval. ^ Indicates a tendency for a difference (> 0.05 *p* < 0.10).

**Figure 3 animals-12-00170-f003:**
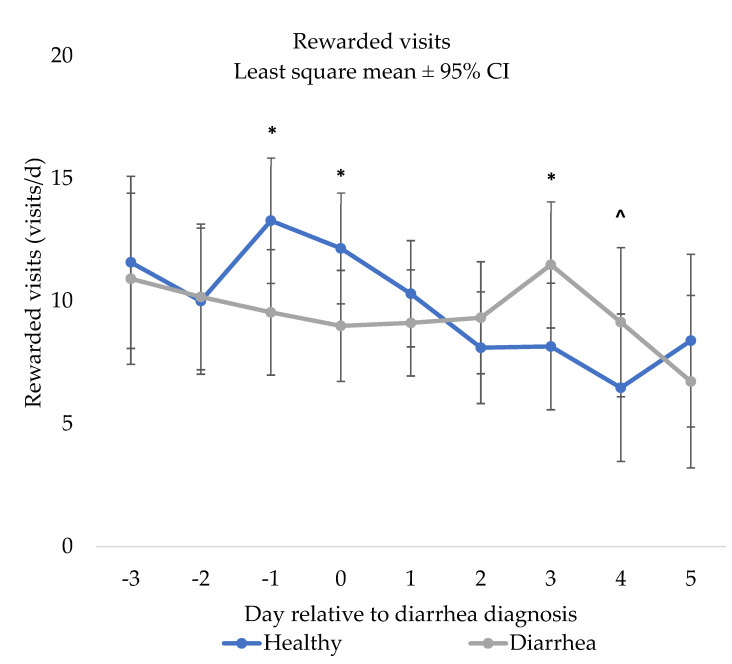
Results of the repeated measures mixed linear regression model assessing the relationship between day relative to neonatal calf diarrhea diagnosis (day 0) and least square mean rewarded visits (visits/d) of pair matched control (*n* = 31) and diarrhea (*n* = 31) calves fed by an automated milk feeder, preweaning. A calf was defined as having NCD if they had a fecal consistency score of 2 or 3 (runny to watery consistency) for two or more consecutive days. Calves were matched on farm, gender and age and the model included calf and pair number to account for the matched pair analysis. The error bars represent ±95% confidence interval. * Indicates a significant difference (*p* ≤ 0.05), and ^ indicates a tendency (> 0.05 *p* < 0.10).

**Figure 4 animals-12-00170-f004:**
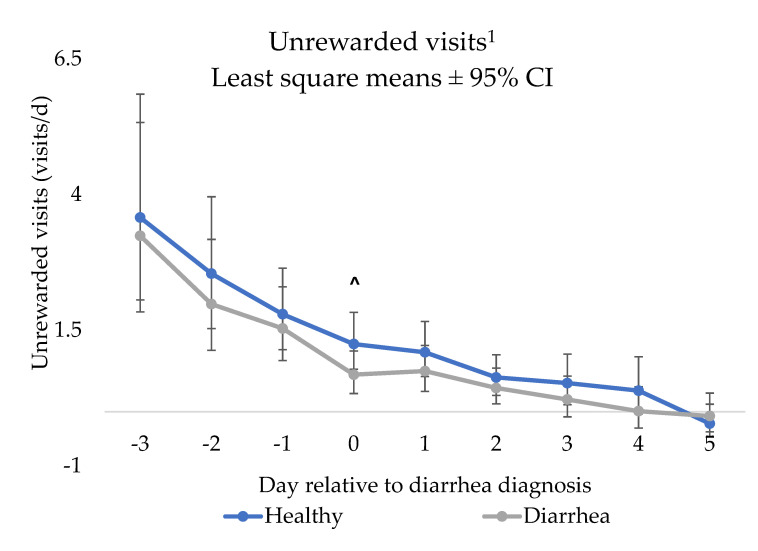
Results of the repeated measures mixed linear regression model assessing the relationship between day relative to neonatal calf diarrhea diagnosis (day 0) and least square mean unrewarded visits (visits/d) ^1^ of pair matched control (*n* = 31) and diarrhea (*n* = 31) calves fed by an automated milk feeder, preweaning. A calf was defined as having NCD if they had a fecal consistency score of 2 or 3 (runny to watery consistency) for two or more consecutive days. Calves were matched on farm, gender and age and the model included calf and pair number to account for the matched pair analysis. ^1^ Unrewarded visits were non-normally distributed, significant differences (*p* ≤ 0.05) were obtained by transformation with common log and a correction factor. The least square means and error bars represent ±95% confidence interval of the back transformed predicted values. ^ indicates a tendency (> 0.05 *p* < 0.10).

**Table 1 animals-12-00170-t001:** Baseline characteristics of case and control calves where control calves (*n* = 31) did not have neonatal calf diarrhea (NCD), and case calves (*n* = 31) did. Cases of NCD were defined as having a fecal consistency score of 2 or 3 (e.g., runny to watery consistency) for two or more days. Cases calves were matched to control calves based on farm, sex, and age. All calves incurred NCD preweaning and were fed by automated milk feeder.

Characteristic	Controls ^1^	Cases ^2^
Female	72%	72%
Mean age (days)	13 ± 4	12 ± 5
Mean days on automated feeder	8 ± 4	8 ± 4

^1^ Controls = calves without neonatal calf diarrhea (NCD); ^2^ Cases = sick calves with NCD.

**Table 2 animals-12-00170-t002:** The association of diarrhea status (healthy or diarrhea) with feeding behaviors of dairy calves (*n* = 31 matched pairs ^1^) in the 3 days before and the 5 days after diarrhea diagnosis. Feeding behaviors (average drinking speed, total milk consumption, number of rewarded and unrewarded visits ^2^) were recorded by an automated milk feeder from which calves were fed milk replacer ad libitum. Results are reported as least squares means and the 95% confidence interval ^3^.

Variable	Controls	Cases	*F*-Value ^4,5^	*p*-Value (Diarrhea Status) ^3^	*p*-Value (Diarrhea Status Interaction by Day)
Milk intake (mL/d)	8465.07 [95% CI 7846.86–9083.23]	7081.29 [95% CI 6463.08–7699.50]	10.45_1,30_	**0.001**	**0.001**
Drinking speed (mL/min)	508.62 [95% CI 450.92–566.33]	481.37 [95% CI 423.66–539.07]	0.47_1,30_	0.50	0.07
Rewarded visits (visits/d)	9.83 [95% CI 8.57–11.08]	9.49 [95% CI 8.23–10.74]	0.15_1,30_	0.70	**0.03**
Unrewarded visits (visits/d)	1.03 [95% CI 0.73–1.38]	0.75 [95% CI 0.49–1.05]	1.79_1,30_	0.19	**0.001**

^1^ All calves were pair matched to healthy calves by age at diagnosis, gender, and farm; ^2^ Diarrhea was defined as feces which spread easily and or sifted through bedding for two consecutive days; ^3^ Significance generated from linear mixed models *p* < 0.05; ^4^ non-normally distributed log10 transformation for significance *p* < 0.05, back transformed values are reported for least square means and 95% CI; ^5^ Subscripts refer to numerator and denominator degrees of freedom. The bold indicates statistical significance for easy interpretation for the reader.

**Table 3 animals-12-00170-t003:** Diagnostic abilities of all feeding behaviors which were significantly associated with neonatal calf diarrhea (NCD) status on the day of diagnosis (day 0) for control (*n* = 31) and NCD pair matched preweaned calves (*n* = 31) fed by an automated milk feeder. Cases of NCD were defined as having a fecal consistency score of 2 or 3 (runny to watery consistency) for two or more consecutive days. Cases calves were matched to control calves based on farm, sex, and age. Sensitivity was defined as the probability that a truly diseased calf was identified as diseased using its feeding behaviors collected from the automated milk feeder whereas specificity was defined as the probability that a truly healthy calf was identified as being healthy using feeding behaviors from the automated milk feeder.

Feeding Behavior	Optimal Cutpoint	Sensitivity at Cutpoint	Specificity at Cutpoint	Area under ROC Curve
Milk intake (mL)	6025	0.44	0.28	0.36
Rewarded visits (/d)	11	0.34	0.69	0.52

## Data Availability

This dataset is available for open access in a Mendeley repository doi:10.17632/djf4fkgp5w.1.

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
