# Peer review of "Associations between Feeding Behaviors Collected from an Automated Milk Feeder and Neonatal Calf Diarrhea in Group Housed Dairy Calves: A Case-Control Study"

_animals, 2022, doi:10.3390/ani12020170_

Round 1
Reviewer 1 Report
Please see attached file.

Author Response
Reviewer 1
Au: We thank you for your time and service to our industry with reviewing this manuscript. We have addressed all of your suggestions including adding the number of calves by gender to the manuscript and acknowledging limitations in our discussion such as not having birthweight as a covariate to explore in this convenience sample analysis. We have included all changes in the manuscript in the letter as well. We thank you for your time and hope this manuscript is now acceptable for publication.
L101 Provide the breakdown of male to female
Au: We added the gender breakdown (62 calves of which 16 were bulls and 46 were heifers) to line 244
Awkward sentence, reword lines 157-159
Au: We have rephrased and it now reads… “We also required the feeding behavior data to be complete for all calves enrolled on this study from days -3 to day 5 after NCD diagnosis.” Line 162-163
L171 It may be better to use farm or year as a covariate or alternatively sire or dam as a random effect as calves in year one could be half-sibs or full-sibs of calves in year two. This may be a moot point if calves were unrelated. Simply adding in a fixed effect of year may not be sufficient to tease out these environmental relationships.
Au: Data were collected from January to July 2017 and January to November of 2018 at the University of Guelph Livestock Research and Innovation Centre – Elora Dairy Facility (Elora, ON, Canada) and from June to August of 2020 at the Grober Young Animal Development Center (Cambridge, ON, Canada). In lines 95-99. We matched the case control pairs by farm and by year to eliminate the farm/year effect. Unfortunately, calves at the Grober facility were sourced from multiple locations, thus we cannot account for genetic relationships between the calves.
L194 Should consider incorporating calf birth weight as a covariate.
Au: Thank you for your comment, unfortunately calves from the Grober facility were sourced from multiple farms and we do not have access to calf birth weight to include this as a covariate. We have added this as a limitation to our limitations section of the discussion. It now reads… “Furthermore, this data was collected as part of a convenience sample which may have reduced the external validity of the study. For example, since calves from the Grober facility were sourced from multiple locations, we did not have serum total protein status, or birthweights to include as a covariate in the model.” In lines 469-471
Figure 3 Line 310, odd spacing
Au: Corrected
Reviewer 2 Report
General comments to the author
The manuscript “Association between feeding behaviors collected from an automated milk feeder and neonatal calf diarrhea in group housed dairy calves: a case control study” is an interesting and well written paper to get earlier information about sick calves on the example of calf diarrhea via KI and data out of an automated milk feeder.
Before publication I have the following minor comments.
Comments
Ln 109-125: Please give more information and describe more precisely information about the colostrum management at Elora Dairy facility and if it possible about the animals from the Grober Young Animal Development Center which is hardly possible? How much animals were placed per feeding station at each AMF during the observation?
Ln 131-135: Did you measure total blood plasma protein content of the calves at arrival at the Grober Young Animal Development Center to check the immune status of the calves?
Ln 251: Please change into sick calves with NCD and not sicks.
Ln 340-351: The resulted sensitivity of 69 % and the specificity of 22 % is low. If you look on the sensitivity and the specificity calculated from milk intake and rewarded visits (see table 3) are dissatisfied low. What do we need in future research trials to increase the sensitivity and specificity to use data out of the AMF to identify sick animals which have NCD? You discussed that also in your conclusion. Do we need more cases (truly sick and truly healthy calves) or do we need much more animals in total to increase the sensitivity and the specificity? Or do we need more complex algorithms which combine more parameter together of feeding behavior like drinking speed, milk intake and rewarded visits to improve the information from the AMF?
Author Response
Reviewer 2
Au: We thank you for your time and service to our industry with reviewing this manuscript. We have addressed all of your suggestions including adding colostrum management of these calves to the manuscript and acknowledging limitations in our discussion such as not having total serum protein status at arrival to explore as a covariate in this convenience sample analysis. We have included all of our changes to the manuscript in the letter for clarity. We thank you for your time and hope this manuscript is now acceptable for publication.
Ln 109-125: Please give more information and describe more precisely information about the colostrum management at Elora Dairy facility and if it possible about the animals from the Grober Young Animal Development Center which is hardly possible? How much animals were placed per feeding station at each AMF during the observation? Au: We have added information regarding the colostrum management of these facilities to the materials and methods. It now reads…“In brief, calves at the Elora Dairy Facility were fed 3 L of colostrum by bottle within 2 hours and offered another 3 L 12 hours later [24]. In lines 105-108. Since Grober Young Animal Development Center was an open herd, colostrum management information for these calves was not available.” In lines 104-107. Both facilities used “dynamic flow” and there were “10-15 calves at the AMF in the Elora facility” and “7 calves per pen at the Grober facility” in lines 118-121.
Ln 131-135: Did you measure total blood plasma protein content of the calves at arrival at the Grober Young Animal Development Center to check the immune status of the calves? Au: Unfortunately we do not have the serum total protein status for the calves housed at the Grober facility since this was a convenience sample study. We have added this as a limitation to this study. It now reads…“Furthermore, this data was collected as part of a convenience sample which may have reduced the external validity of the study. For example, since calves from the Grober facility were sourced from multiple locations, we did not have serum total protein status or birthweights to include as a covariate in the models” In lines 469-471
Ln 251: Please change into sick calves with NCD and not sicks. Au: Corrected in line 252
Ln 340-351: The resulted sensitivity of 69 % and the specificity of 22 % is low. If you look on the sensitivity and the specificity calculated from milk intake and rewarded visits (see table 3) are dissatisfied low. What do we need in future research trials to increase the sensitivity and specificity to use data out of the AMF to identify sick animals which have NCD? You discussed that also in your conclusion. Do we need more cases (truly sick and truly healthy calves) or do we need much more animals in total to increase the sensitivity and the specificity? Or do we need more complex algorithms which combine more parameter together of feeding behavior like drinking speed, milk intake and rewarded visits to improve the information from the AMF? Au: We agree that we should include this. It now reads.. “ Based on the literature, we believe that more complex algorithms are needed to increase the sensitivity and specificity of disease detection for NCD in calves, because the time between the pathogen exposure, incubation period and symptoms of NCD is around 24-48 hours in calves [21,22]. It is possible that multiple behaviors (i.e., activity levels with feeding behavior) may increase the sensitivity and specificity of an algorithm’s ability to detect disease in calves in a timely manner. Alternatively, it is also possible that with a larger population of dairy calves, machine learning techniques can tease out feeding behavior differences associated with NCD without incorporating multiple precision technologies. Thus, more research is needed to develop methods which increase the sensitivity and the specificity of disease detection using feeding behavior in calves.” In lines 490-495